# Emissions of methane in Europe inferred by total column measurements

Debra Wunch[1], Dylan B. A. Jones[1], Geoffrey C. Toon[2], Nicholas M. Deutscher[3,4], Frank Hase[5], Justus Notholt[4], Ralf Sussmann[6], Thorsten Warneke[4], Jeroen Kuenen[7], Hugo Denier van der Gon[7], Jenny A. Fisher[3], and Joannes D. Maasakkers[8]

[1]Department of Physics, University of Toronto, Canada
[2]Jet Propulsion Laboratory, California Institute of Technology, USA
[3]Centre for Atmospheric Chemistry, University of Wollongong, Australia
[4]Institute of Environmental Physics, University of Bremen, Germany
[5]Karlsruhe Institute of Technology, IMK-ASF, Karlsruhe, Germany
[6]Karlsruhe Institute of Technology, IMK-IFU, Garmisch-Partenkirchen, Germany
[7]TNO Dept Climate, Air and Sustainability, Utrecht, The Netherlands
[8]Harvard University, Cambridge, Massachusetts, United States

*Correspondence to:* Debra Wunch (dwunch@atmosp.physics.utoronto.ca)

**Abstract.** Using five long-running ground-based atmospheric observatories in Europe, we demonstrate the utility of long-term, stationary, ground-based measurements of atmospheric total columns for verifying annual methane emission inventories. Our results indicate that the methane emissions for the region in Europe between Orléans, Bremen, Białystok, and Garmisch are overestimated by the state-of-the-art inventories Emission Database for Global Atmospheric Research (EDGAR) v4.2 FT2010
5 and TNO-MACC_III, possibly due to the disaggregation of emissions onto a spatial grid. Uncertainties in the carbon monoxide inventories used to compute the methane emissions contribute to the discrepancy between our inferred emissions and those from the inventories.

## 1 Introduction

Recent global policy agreements have led to renewed efforts to reduce greenhouse gas emissions to cap global temperature
10 rise (e.g., Conference of the Parties-21 (COP-21; UNFCCC, 2015) and the Covenant of Mayors (European Commission, 2016)). This, in turn, has motivated countries to seek methods of reducing their greenhouse gas emissions. In Europe, methane emissions account for a significant fraction (about 11% by mass of $CO_2$ equivalent) of the total greenhouse gas emissions (UNFCCC, 2017). The lifetime of atmospheric methane is significantly shorter than for carbon dioxide, its 100-year global warming potential is significantly larger, and it is in near steady-state in the atmosphere, therefore significant reductions in
15 methane emissions are an effective *short-term* strategy for reducing greenhouse gas emissions (Dlugokencky et al., 2011). Emission reduction strategies that include both methane emission reductions and carbon dioxide reductions are thought to be among the most effective at slowing the increase in global temperatures (Shoemaker et al., 2013). Thus, it is important to know exactly how much methane is being emitted, and the geographic and temporal source of the emissions. This requires

an approach that combines state-of-the-art emissions inventories that contain information about the specific point and area sources of the known emissions, and timely and long-term measurements of greenhouse gases in the atmosphere to verify that the emissions reduction targets are met.

Because atmospheric methane is well-mixed and has a lifetime of about 12 years (Stocker et al., 2013), it is transported far from its emission source, making source attribution efforts challenging from atmospheric measurements alone. Atmospheric measurements are often assimilated into "flux inversion" models to locate the sources of the emissions (e.g., Houweling et al., 2014) but rely on model wind fields to drive transport, and tend to have spatial resolutions that do not resolve sub-regional scales. Methane measurement schemes that constrain emissions on local and regional scales are thus important to help identify the sources of the emissions and to verify inventory analyses. Regional or country-scale emissions are important to public policy as those emissions are reported annually to the United Nations Framework Convention on Climate Change (UNFCCC).

The atmospheric measurement techniques that are used to estimate methane emissions include measurements made in situ, either on the ground, from tall towers, or from aircraft. Remote sensing techniques are also used, either from space or from the ground. The spatial scale of the sensitivity to emissions differs by the measurement technique: surface in situ measurements provide information about local emissions on urban scales (e.g., McKain et al., 2015; Hopkins et al., 2016), aircraft in situ measurements can provide information about regional and synoptic-scale fluxes (e.g., Jacob et al., 2003; Kort et al., 2008, 2010; Wofsy, 2011; Baker et al., 2012; Frankenberg et al., 2016; Karion et al., 2016). Satellite remote sensing techniques provide information useful for extracting emissions information on larger scales (regional to global) (e.g., Silva et al., 2013; Schneising et al., 2014; Alexe et al., 2015; Turner et al., 2015), and for large point or urban sources (e.g., Kort et al., 2012, 2014; Nassar et al., 2017). Several studies have shown the importance of simultaneous measurements of co-emitted species (e.g., $C_2H_6$ and $CH_4$; CO and $CO_2$; Aydin et al., 2011; Simpson et al., 2012; Peischl et al., 2013; Silva et al., 2013; Hausmann et al., 2016; Wunch et al., 2016; Jeong et al., 2017) or co-located measurements (e.g., Wunch et al., 2009, 2016) showing the added analytical power of the combination of atmospheric tracer information. Ground-based remote sensing instruments have been used to estimate methane emissions on urban (e.g., Wunch et al., 2009; Hase et al., 2015; Wunch et al., 2016) and sub-urban (e.g., Chen et al., 2016; Viatte et al., 2017) scales. In Hase et al. (2015), Viatte et al. (2017), and Chen et al. (2016), the authors have placed mobile ground-based remote sensing instruments around a particular emitter of interest (e.g., a city, dairy, or neighbourhood) and have designed short-term campaigns to measure the difference between upwind and downwind atmospheric methane abundances. From these differences, the authors have computed emission fluxes. However, there is a network of non-mobile ground-based remote sensing instruments that have been collecting long-term measurements of atmospheric greenhouse gas abundances. These instruments were not placed intentionally around an emitter of interest, but collectively, they ought to contain information about nearby emissions. To date, there have been no studies that have attempted to extract regional methane emissions information from these existing ground-based remote sensing observatories.

In this paper, we will describe our methods for computing the emissions of methane using five stationary ground-based remote sensing instruments located in Europe in §2. Our results, and comparisons to the state-of-the-art inventories are shown in §3, and we summarize our results in §4.

## 2   Methods

Our study area is the region between five long-running atmospheric observatories situated in Europe. Three of the stations are in Germany: Bremen (Notholt et al., 2014), Karlsruhe (Hase et al., 2014), and Garmisch (Sussmann and Rettinger, 2014). The other two are in Poland (Białystok, Deutscher et al., 2014), and France (Orléans, Warneke et al., 2014). Each station measures

the vertical column-averaged dry-air mole fraction of carbon dioxide ($X_{CO_2}$), carbon monoxide ($X_{CO}$), methane ($X_{CH_4}$), and other trace gas species. The locations are shown in Figure 1, overlaid on a night lights image from the National Aeronautics and Space Administration (NASA) to provide a sense of the population density of the area. These observatories are part of the Total Carbon Column Observing Network (TCCON, Wunch et al., 2011), and have been tied to the World Meteorological Organization trace-gas scale through comparisons with vertically integrated, calibrated in situ profiles over the observatories

(Wunch et al., 2010; Messerschmidt et al., 2011; Geibel et al., 2012; Wunch et al., 2015).

Following a similar method to Wunch et al. (2009, 2016), we estimate emissions of methane from the data recorded from the TCCON observatories, coupled with gridded inventories of carbon monoxide within the region. We compute changes (or "anomalies") in $X_{CH_4}$ and $X_{CO}$ that we will refer to as $\Delta X_{CH_4}$ and $\Delta X_{CO}$, and then compute the slopes relating $\Delta X_{CH_4}$ to $\Delta X_{CO}$. From the computed slopes ($\alpha$), we can infer emissions of methane ($E_{CH_4}$) if emissions of carbon monoxide ($E_{CO}$, in

mass per unit time) are known, using the following relationship:

$$E_{CH_4} = \alpha \frac{m_{CH_4}}{m_{CO}} E_{CO} \tag{1}$$

where $\frac{m_{CH_4}}{m_{CO}}$ is the ratio of the molecular masses of $CH_4$ and CO.

In the Wunch et al. (2009, 2016) papers, measurements from a single atmospheric observatory were used to infer emissions, because the unique dynamics of the region advected the polluted airmass into and out of the study area diurnally. In this paper,

we rely on several stations to provide measurements of the boundary of the study region to measure CO and $CH_4$ emitted between the stations. This analysis relies on a few assumptions about the nature of the emissions. First, that the lifetimes of the gases of interest are longer than the transport time within the region. This is the case both for methane, which has an atmospheric lifetime of 12 years, and for carbon monoxide, which has an atmospheric lifetime of a few weeks. Second, we assume that typical emissions are consistent over time periods longer than a few days so that they are advected together. The

nature of the emissions in this region (mostly residential and industrial energy needs) supports this assumption. Third, we assume that the spatial distribution of the emissions is similar for $CH_4$ and CO, as confirmed by the inventory maps (Fig. A3). This method does not require that carbon monoxide and methane are co-emitted (as they generally do not have the same emissions sources).

To compute anomalies and slopes, we first filter the data to minimize the impacts of data sparsity and air mass differences

between stations (Appendix A). Then, for each station, the daily median value is subtracted from each measurement. This reduces the impact of the station altitude and any background seasonal cycle from aliasing into the results. Subsequently, we compute the differences in the $X_{CH_4}$ and $X_{CO}$ abundances measured at the same solar zenith and solar azimuth angles on the same day at two TCCON stations. By computing anomalies at the same solar zenith angles, we minimize any impact that airmass-dependent biases could have on the calculated anomalies. This analysis is repeated for all combinations of pairs of

stations within the study area. The vertical sensitivity of the TCCON measurements is explicitly taken into account by dividing the anomalies by the surface layer column averaging kernel value, as we assume that the anomalies are due to emissions near the surface. The slopes computed for each year for each pair of stations are shown in Figure 2.

The farthest distance between the European TCCON stations included in this study is between Orléans and Białystok (1580 km). Climatological annual mean surface wind speeds from the National Centers for Environmental Prediction (NCEP)/ National Center for Atmospheric Research (NCAR) reanalysis (Kalnay et al., 1996) within the study area are about $6 \, \text{km} \cdot \text{h}^{-1}$ (Fig. A1). The air from Orléans will quickly mix vertically from the surface where the winds aloft are more rapid than at the surface (see Appendix B). Thus, air from Orléans would normally reach Białystok in a few days. To determine whether these anomalies are consistent throughout the transport time through the study area, we compute anomalies between sites lagged by up to 14 days. The slopes of the anomalies do not change significantly or systematically with the lag time (Appendix B; Fig. A2), presumably because the atmospheric composition within the study area is relatively well-mixed or because the emissions are relatively consistent from day to day within the study area.

Previous papers have used carbon dioxide instead of carbon monoxide to infer methane emissions. We choose to compute emissions using measurements of $X_{CO}$ instead of $X_{CO_2}$ in this work because the natural $CO_2$ fluxes in the region are large compared with the anthropogenic emissions, and they have a strong diurnal and seasonal cycle. The distance between the stations is large enough that local (sub-daily) uptake of $CO_2$ differs from station to station, significantly obscuring the relationships between methane and carbon dioxide, and thus the anomaly slopes, especially in the summer months. While the emissions inventory of anthropogenic $CO_2$ may be more accurate than the CO inventory in the region, the presence of these large natural fluxes of $CO_2$ precludes its use in the anomaly slope calculation. The accuracy of our method, then, is limited by the accuracy of the carbon monoxide emission inventory. Fires could provide a large flux of CO without a large $CH_4$ flux, and this should also be taken into consideration in these types of analyses. In our study area, fluxes from fires are small.

## 2.1 Inventories

To obtain an estimate of carbon monoxide emissions ($E_{CO}$) within the study area, we use gridded inventories, and sum the emissions within the study area to compare with our emissions inferred from the TCCON measurements (see Appendix C and Fig. A3 for details). The two inventories employed here are the Emission Database for Global Atmospheric Research (EDGAR) and TNO-MACC_III. The EDGAR version v4.3.1_v2 of January 2016 annual gridded inventory is available at $0.1° \times 0.1°$ spatial resolution and reports global emissions from the year 2000 to 2010 (Olivier et al., 1994; EC-JRC and PBL, 2016). The TNO-MACC_III inventory is a Europe-specific air quality emissions inventory, available on a $0.125° \times 0.0625°$ grid, and reports emissions for 2000–2011 (Kuenen et al., 2014). Both EDGAR and TNO-MACC_III provide spatially and temporally coincident methane inventories which we use to compare with our inferred emissions. We use the EDGAR version v4.2 FT2010 and the TNO-MACC_III methane inventories.

Using country-level emissions reported through 2015 from the European Environment Agency (EEA, 2015), we extrapolate the EDGAR and TNO-MACC gridded inventory CO emissions for the study area through 2015. This facilitates more direct comparisons with the TCCON measurements, which begin with sufficient data for our study in 2009. We extrapolate the

emissions by scaling the total emissions from the countries that are intersected by the area of interest (Germany, Poland, Belgium, France, Luxembourg, Czech Republic) to the last reported year of emissions from the inventory. We then assume that the same scaling factor applies for each subsequent year. The details of the extrapolation method are in Appendix D and Figs. A4 and A5.

The time series of the reported emissions from 2000–2015 are shown in Fig. 3. The inventories and scaled country-level reported emissions for this region suggest that emissions of CO and $CH_4$ have decreased by about 40% and 20%, respectively, between 2000 and 2015. The TNO-MACC_III carbon monoxide emissions are on average 15% higher than the EDGAR v.4.3.1 emissions in the study area. The total TNO-MACC_III and EDGAR methane emissions agree to within 2% in the study area.

An earlier version of the EDGAR carbon monoxide inventory was evaluated by Stavrakou and Müller (2006) and Fortems-Cheiney et al. (2009), who assimilated satellite measurements of CO using the EDGAR v3.3FT2000 CO emissions inventory as the a priori. Stavrakou and Müller (2006) found that over Europe, the a posteriori emissions increase by less than 15% when assimilating carbon monoxide from the Measurements of Pollution in the Troposphere (MOPITT) satellite instrument (Emmons et al., 2004). Fortems-Cheiney et al. (2009) assimilated Infrared Atmospheric Sounding Interferometer (IASI) CO (Clerbaux et al., 2009) and MOPITT CO, and found that the a posteriori emissions increase by 16% and 45%, respectively.

The more recent EDGAR v4.3.1 CO emissions in our study are 24% lower than the EDGAR v3.3FT2000 CO emissions for the year 2000, so it may be that the EDGAR v4.3.1 CO emissions are significantly underestimated. However, assimilations of CO are known to be very sensitive to the chemistry described in the model: most notably the OH chemistry (Protonotariou et al., 2010; Yin et al., 2015). Therefore it is difficult to determine how much of the discrepancy between versions of the model is from the inventory or the model chemistry.

The EDGAR methane inventory has been evaluated in several previous studies. It has been shown to overestimate regional $CH_4$ emissions (e.g., Wunch et al., 2009; Wecht et al., 2014), but to underestimate oil and gas emissions (e.g., Miller et al., 2013; Buchwitz et al., 2017). However, recent methane isotope analysis by Röckmann et al. (2016) has suggested that the EDGAR inventory overestimates fossil fuel-related emissions. The study area of interest here has little oil and gas production, except for some test sites in Poland (USEIA, 2015), no commercial shale gas industry, and few pipelines.

## 2.2 Model Experiment

To test whether the anomaly method described in §2 can accurately infer methane emissions, we conducted a modeling experiment using version v12.1.0 of the GEOS-Chem model (www.geos-chem.org) to simulate methane and carbon monoxide for the year 2010. The model is driven by the Modern-Era Retrospective analysis for Research and Applications, Version 2 (MERRA-2) meteorology from the NASA Global Modeling and Assimilation Office. The native resolution of the meteorological fields is 0.25°x 0.3125°, with 72 vertical levels from the surface to 0.01 hPa, which we degraded to 2°x 2.5°and 47 vertical levels. We use the linear CO-only and $CH_4$-only simulations of GEOS-Chem, with prescribed monthly mean OH fields. In the CO-only simulation, global anthropogenic emissions are from EDGAR v4.3.1, which are overwritten regionally with the following emissions: the Cooperative Programme for Monitoring and Evaluation of the Long-range Transmission of Air Pollutants in Europe (EMEP), the U.S. Environmental Protection Agency National Emission Inventory for 2011 (NEI2011), the MIX inven-

tory for Asia, the Visibility Observational (BRAVO) Study Emissions Inventory for Mexico, and the criteria air contaminants (CAC) inventory for Canada. The sources of CO from the oxidation of $CH_4$ and volatile organic compounds (VOCs) are prescribed following Fisher et al. (2017). For the $CH_4$-only simulation, the emissions are as described in Maasakkers et al. (2019). Global anthropogenic emissions are from EDGAR v4.3.2, but the U.S. emissions were replaced with those from Maasakkers

et al. (2016), and emissions from wetlands are from WetCHARTs version 1.0 (Bloom et al., 2017). For both CO and $CH_4$ simulations, emissions from biomass burning are from the Quick Fire Emissions Dataset (QFED) (Darmenov and Silva, 2015). The biomass burning in the study area produces less than 2% of the total anthropogenic emissions of CO.

    We used identical OH fields (from version v7-02-03 of GEOS-Chem) for the CO and $CH_4$ simulations so that the chemical losses of methane and carbon monoxide are consistent, and ran tagged CO experiments so that we could identify the source of

the emissions. The model atmospheric carbon monoxide and methane profiles were integrated to compute simulated $X_{CO}$ and $X_{CH_4}$. To illustrate the sensitivity of the modeled fields to European emissions, we show in Fig. 4 the seasonal means of the modeled $X_{CO}$ sampled at the five TCCON stations. Also plotted is the column contribution ($X_{CO-Eur}$) from CO emissions only in Europe (defined as the broader region between $0°E$ - $45°E$ and $45°N$ - $55°N$). As can be seen, the spatial pattern of the differences in modeled $X_{CO}$ between the TCCON stations are reflected in $X_{CO-Eur}$. We calculated the anomalies in $X_{CO}$ and

$X_{CO-Eur}$, using the same approach employed with the atmospheric data, and found that the anomalies in $X_{CO-Eur}$, which represent the direct influence of European emissions on atmospheric CO, account for about 35% of the anomalies in $X_{CO}$. This confirms that the $X_{CO}$ anomalies between the TCCON stations are sensitive to European emissions.

    To estimate the modeled $CH_4$ emissions using the modeled CO, the modeled $X_{CO}$ and $X_{CH_4}$ were interpolated to the locations of the TCCON stations and anomalies and slopes were computed. We then applied Equation 1 to our anomaly slopes

to compute methane emissions from the known CO emissions, accounting for only the CO emissions from anthropogenic, biomass burning, and biofuel sources. We neglect sources of CO emissions from the oxidation of $CH_4$ and VOCs because the column enhancements for those emissions are relatively spatially uniform across this region of Europe and thus should not contribute significantly to the anomalies. The resulting annual $CH_4$ emissions agree well with the model emissions: the inferred emissions from the anomaly analysis are higher than the model emissions by less than 2% percent (Fig. 5).

While the inferred annual emissions agree well with the modeled annual emissions, the seasonal pattern of the emissions inferred from the anomaly analysis differs from that of the model. The anomaly analysis overestimates emissions in the winter and underestimates emissions in the summer. This may be due to small spatial inhomogeneities in the column enhancements from VOC (biogenic) emissions that influence the anomaly analysis most in summertime when VOC emissions are largest. Including the VOC emissions in the total carbon monoxide emissions leads us to infer annual methane emissions that are

overestimated by 15%, increasing the inferred summertime emissions without significantly changing the inferred wintertime emissions.

    The seasonal analysis suggests that the 2% agreement in the annual emission estimate may reflect the compensating effects of discrepancies over the seasonal cycle, and improving the seasonal estimate may require a better treatment of the VOC contribution to atmospheric CO. Nevertheless, the results here suggest that for this region of Europe, where VOC and methane

oxidation emissions lead to relatively spatially uniform column enhancements, and fire emissions are small, we can successfully use the anomaly method described in §2 to infer annual methane emissions.

## 3 Results and Discussion

To compute methane emissions, we apply equation 1 to our anomaly slopes and the inventory-reported carbon monoxide emissions in the study region (Fig. 6). If we choose the mean of the reported CO emissions from EDGAR v4.3.1 and TNO-MACC_III, the methane emissions we compute within the study area based on the TCCON measurements are $1.7 \pm 0.3 \, \mathrm{Tg \cdot yr^{-1}}$ in 2009, with a non-monotonic decrease to $1.2 \pm 0.3 \, \mathrm{Tg \cdot yr^{-1}}$ in 2015. The uncertainties quoted here are from the standard errors on the data slope fitting only; we have not included uncertainties from the inventories. The magnitude of methane emissions we compute from the TCCON data are, on average, about 2.3 times lower than the methane emissions reported by EDGAR, and about 2 times lower than the methane emissions reported by TNO-MACC_III.

Our method of inferring methane emissions depends critically on the carbon monoxide inventory. The carbon monoxide emissions for 2010 in the study area from our GEOS-Chem model run, derived from EMEP emissions, were 6.4 Tg, about 35% higher than the average of the EDGAR and TNO-MACC_III emissions for that year. This magnitude underestimate has also been suggested by Stavrakou and Müller (2006) and Fortems-Cheiney et al. (2009) using independent data. Using the GEOS-Chem carbon monoxide emissions increases the methane emissions inferred by the anomaly analysis to $2.4 \pm 0.3$ Tg in 2010. This value remains lower than the EDGAR and TNO-MACC_III methane emissions estimates for 2010, which are 3 Tg, but by only 20%. Therefore, we find that the inventories likely overestimate methane emissions, but the accuracy of our results relies on the accuracy of the carbon monoxide inventory.

Although the EDGAR and TNO-MACC_III inventories agree to within 15% in carbon monoxide emissions and 2% in methane emissions in the study region, they spatially distribute these emissions differently. Maps of the spatial differences between the TNO-MACC_III and EDGAR emissions are shown in Fig. 7 for carbon monoxide and Fig. 8 for methane. EDGAR estimates larger emissions of carbon monoxide from the main cities in the study region and the surrounding areas. This is clearly visible from the difference map (Fig. 7), where cities such as Hamburg, Berlin, Prague, Wrocław, Warsaw, Munich, Paris, and Vienna appear in blue. However, the overall carbon monoxide emissions from TNO-MACC_III in the study area are higher than EDGAR, and this comes from regions between the main cities, particularly in Poland and eastern France.

The differences between EDGAR and TNO-MACC_III methane emissions also show that the EDGAR emissions estimates near large cities are significantly larger (Fig. 8). In contrast to the carbon monoxide spatial distribution, the TNO-MACC_III methane emissions are generally smaller everywhere, except for discrete point sources.

Comparing 2010 reported country-level carbon monoxide emissions with the inventories shows reasonable agreement, which is expected since the inventories use country-level reports as input. The sum of the carbon monoxide emissions within the entire countries of Germany, Poland, France, Luxembourg, Belgium, and Czech Republic differ between EDGAR and TNO-MACC_III by 18%, with EDGAR estimates lower than those from TNO-MACC_III. Emissions from Germany, most of which are included in the study area, differ by only 6% between EDGAR and TNO-MACC_III, again with EDGAR estimates lower

than TNO-MACC_III. The national carbon monoxide emissions reported to the Convention on Long-range Transboundary Air Pollution (LRTAP Convention, https://www.eea.europa.eu/ds_resolveuid/0156b7a0ca47485593e7754c52c24afd, EEA, 2015) agree to within a few percent of the TNO-MACC_III country-level emissions (e.g., 5.5% for Germany in 2010).

The differences between 2010 country-level emissions estimates are larger for methane: EDGAR estimates are larger than TNO-MACC_III estimates by 36% when summing all countries intersected by the study area, and 8% when considering only German emissions. The TNO-MACC_III country-level emissions estimates agree to within a few percent of the UNFCCC (http://di.unfccc.int/time_series) country-level reported methane emissions (e.g., 8% for Germany in 2010).

The differences between the EDGAR and TNO-MACC_III inventories suggest that the spatial distribution of emissions is less certain than the larger-scale emissions, since the total carbon monoxide and methane emissions between the inventories agree to within 15% and 2% respectively in the study area, but these estimates can disagree by a factor of two on city scales.

If we assume that the country-scale methane emissions are correctly reported in EDGAR and TNO-MACC_III, our results indicate that the methane emissions in the region are incorrectly spatially distributed in the inventories. It could be that point or urban sources outside the study area, but within the countries intersected by the study area, emit a larger proportion of the country-level emissions than previously thought.

## 4   Conclusions

Using co-located measurements of methane and carbon monoxide from five long-running ground-based atmospheric observing stations, we have shown that in the area of Europe between Orléans, Bremen, Białystok, and Garmisch, the inventories likely overestimate methane emissions, and point to a large uncertainty in the spatial distribution (i.e., the spatial disaggregation) of country-level emissions. However, the magnitude of our inferred methane emissions relies heavily on the EDGAR v4.3.1 and TNO-MACC_III carbon monoxide inventories, and thus there is a need for rigorous validation of the carbon monoxide inventories.

This study demonstrates the potential of clusters of long term ground-based stationary monitoring of total columns of atmospheric greenhouse and tracer gases. It also shows the potential of having co-located measurements of multiple pollutants to derive better estimates of emissions. These types of observing systems can help policymakers verify that greenhouse gas emissions are reducing at a rate necessary to meet regulatory obligations. The atmospheric measurements are agnostic to the source (and country of origin) of the methane, measuring only what is emitted into the atmosphere in a given area. Thus they can help validate and reveal inadequacies in the current inventories, and in particular, how country-wide emission reports are disaggregated on a grid. To enhance these results, simultaneous measurements of complementary atmospheric trace gases, such as ethane, acetylene, nitrous oxide, nitrogen dioxide, ammonia, and isotopes would help distinguish between sources of methane. This would provide additional, valuable information that would likely improve inventory disaggregation.

*Data availability.* TCCON data are available from the TCCON archive, hosted by the California Institute of Technology at https://tccondata. org. The Emission Database for Global Atmospheric Research (EDGAR) inventory is available from the European Commission, Joint Research Centre (JRC) / Netherlands Environmental Assessment Agency (PBL), http://edgar.jrc.ec.europe.eu. The GEOS-Chem v12.1.0 model is available from https://doi.org/10.5281/zenodo.1553349.

## 5  Appendix A:  Filtering

The filtering method was designed to remove days of data for which the atmospheric air mass was inconsistent between sites (e.g., a front was passing through or there were significant stratospheric incursions into the troposphere), and for years in which there were too few simultaneous measurements at a pair of TCCON stations to compute robust annually-representative anomalies.

10    To address the consistency of the air mass between sites, we retained days on which the retrievals of hydrogen fluoride ($X_{HF}$) were between 50 parts-per-trillion (ppt) and 100 ppt, and deviated by less than 10 ppt of the median $X_{HF}$ value for all sites on that day. HF is a trace gas that exists only in the stratosphere, and thus serves as a tracer of tropopause height (Washenfelder et al., 2003; Saad et al., 2014). Since the concentration of $CH_4$ decreases significantly above the tropopause in the mid-latitudes, its total column dry-air mole fraction ($X_{CH_4}$) is sensitive to the tropopause height. Filtering out days on 15 which $X_{HF}$ varies significantly between sites also ensures that the anomalies (and thus the slopes) are minimally impacted by stratospheric variability. This filter removed less than $5\%$ of the data.

To ensure that the anomalies are representative of the full year, we require that each year has 400 coincident measurements across at least three seasons.

## Appendix B:  Transport time between stations

20    Figure A1 shows the annual change in monthly mean climatological wind speeds from the NCEP/NCAR reanalysis (Kalnay et al., 1996). These are interpolated to surface pressure and 850 hPa pressures (~1500 m geopotential height) from model (sigma) surfaces and cover January 1948 through March 2017. Vertical mixing into the boundary layer occurs on the time scale of a day or two (Jacob, 1999), and thus the relevant wind speed is between the surface and 850 hPa. The annual mean surface wind speed is $6 \; km \cdot hr^{-1}$, which gives a mean transit time between Orléans and Białystok of 11 days. The annual mean 850 25 hPa winds are $17 \; km \cdot hr^{-1}$, which give a shorter mean transit time between Orléans and Białystok of 4 days.

To test whether the transport time impacts the anomalies, we computed the slopes for time lags between sites of $0 - 14$ days. Figure A2 shows a small change in anomaly slope as a function of the lag used to calculate the anomalies. This figure shows that the transport time between TCCON stations is of negligible importance to the slopes and lends weight to the decision to compute anomalies from data recorded at two TCCON stations on the same day.

## Appendix C:  Computing study area emissions from the inventories

The study area emissions for 2010 are shown in Figure A3. We define the study area as the area bounded by the TCCON stations at (clockwise from the West) Orléans, Bremen, Białystok, and Garmisch, which is marked by the black lines in the figure. To compute the emissions from the study area, the grid points intersected by and contained within the solid black lines are summed for each year. The EDGAR v4.3.1_v2 emissions inventory for CO and FT2010 inventory for $CH_4$ provide estimates for years 2000–2010. The TNO-MACC_III inventory provides emissions estimates for both CO and $CH_4$ for years 2000–2011.

## Appendix D:  Projecting inventory emissions beyond 2010

Using data from the European Environment Agency National Database (European Environment Agency, 2016), we extrapolate the inventory CO and $CH_4$ emissions for the study area through 2015. This is done by summing the total emissions for the five countries that are intersected by the study area (France, Belgium, Germany, Poland, Luxembourg, Czech Republic), and normalizing the emissions to the last year of the inventory (2010 for EDGAR, 2011 for TNO-MACC_III). Figures A4 and A5 show the process for the EDGAR and TNO-MACC_III CO and $CH_4$ emissions, respectively.

The top panel of Fig. A4 shows the reported country-level emissions for the years 1990–2015, their sum (black stars), and the sum of the inventory emissions for the years available (2000–2010 for EDGAR; 2000–2011 for TNO-MACC_III) in squares. The second and third panels show the ratio of the country-level emissions to the area emissions, normalized to 1 for the last year available in the inventory. These panels show that the ratio of the summed country total emissions to the emissions from the area of interest is less variable from year to year than the emissions reported for individual countries. Thus, we choose to extrapolate the area emissions using the country total emissions, scaled to the last year of the inventory for the study area.

The bottom panel shows the results of using a single scaling factor to estimate the study area emissions from the country-level emissions for each year. We use the summed study area emissions for the years available, and the extrapolated emissions through 2015 for subsequent analysis (e.g., Figs. 3 and 6).

*Author contributions.*  DW designed the study, performed the analysis, and wrote the paper. DBAJ ran the GEOS-Chem model, supported by the CO and $CH_4$ work of JAF and JDM. JK and HDvdG provided the TNO-MACC_III inventory. GCT helped refine the data analysis methodology. NMD, FH, JN, RS, and TW provided TCCON data. All coauthors read and provided feedback on the contents of the paper and helped interpret the results.

*Competing interests.*  The authors declare no competing interests.

*Acknowledgements.* Emission Database for Global Atmospheric Research (EDGAR) data were obtained from http://edgar.jrc.ec.europe.eu. TCCON data were downloaded from the CaltechDATA archive: https://tccondata.org. The NASA Earth Observatory images were prepared by Joshua Stevens, using Suomi NPP VIIRS data from Miguel Román, NASA's Goddard Space Flight Center. TCCON data were obtained from the TCCON archive, hosted by the California Institute of Technology at https://tccondata.org. The GEOS-Chem v12.1.0 model was obtained from https://doi.org/10.5281/zenodo.1553349. The authors would like to thank two anonymous reviewers for thoughtful comments and suggestions that significantly strengthened the paper.

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

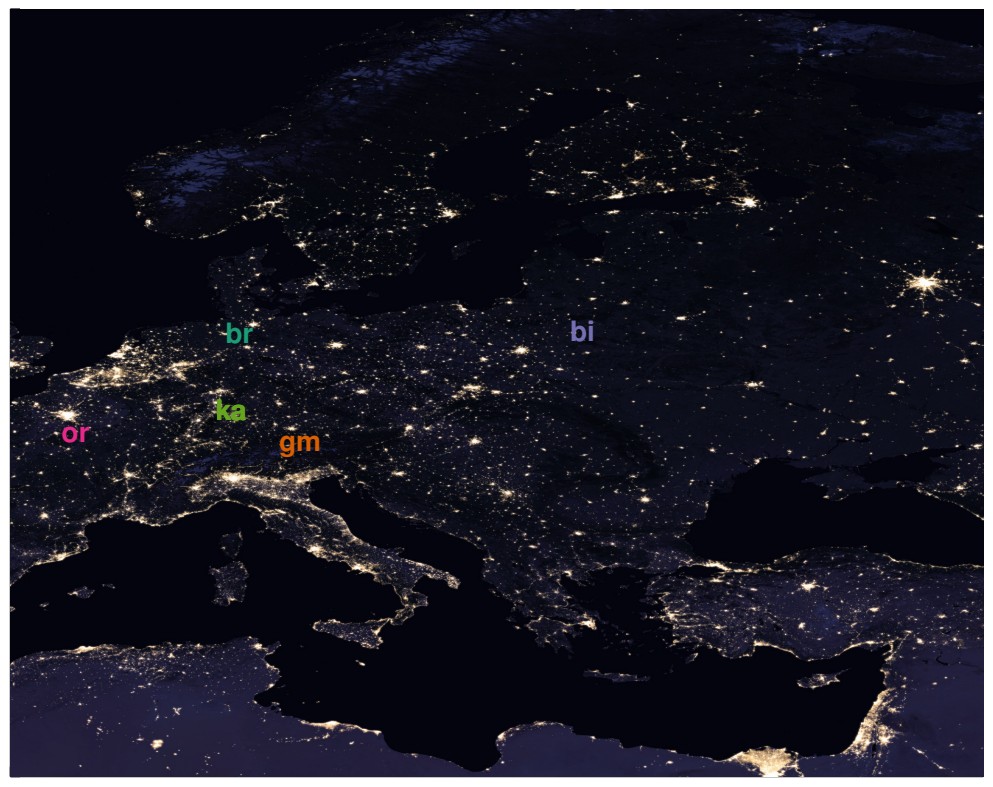

**Figure 1.** The locations of the TCCON observatories overlaid on a NASA night lights image. From west to east, the stations are: Orléans (or, pink), Karlsruhe (ka, green), Bremen (br, blue-green), Garmisch (gm, orange), Białystok (bi, purple).

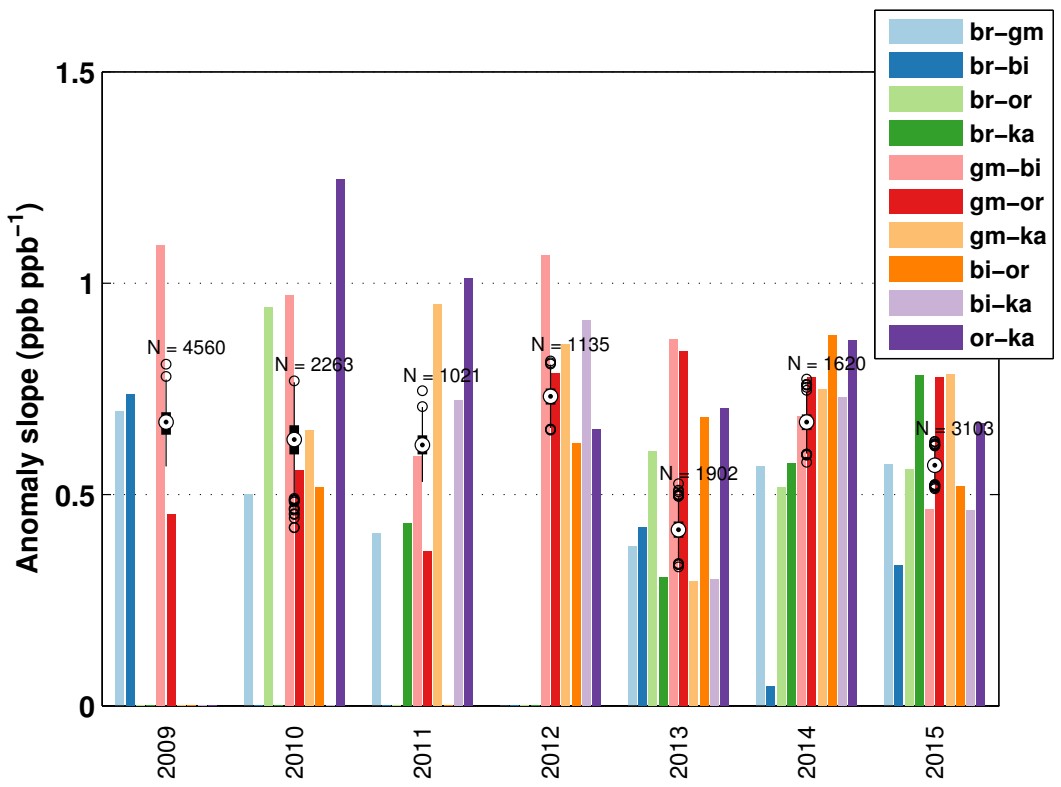

**Figure 2.** The bars show the methane to carbon monoxide anomaly slopes for each site pair. The method of computing these anomaly slopes is detailed in §2 of the main text. The black targets indicate the median value of the slope for that year, when all site pairs are considered simultaneously, and the $25^{th}$ and $75^{th}$ quartiles about that median value are indicated by the vertical black bars. Outliers are indicated by open black circles.

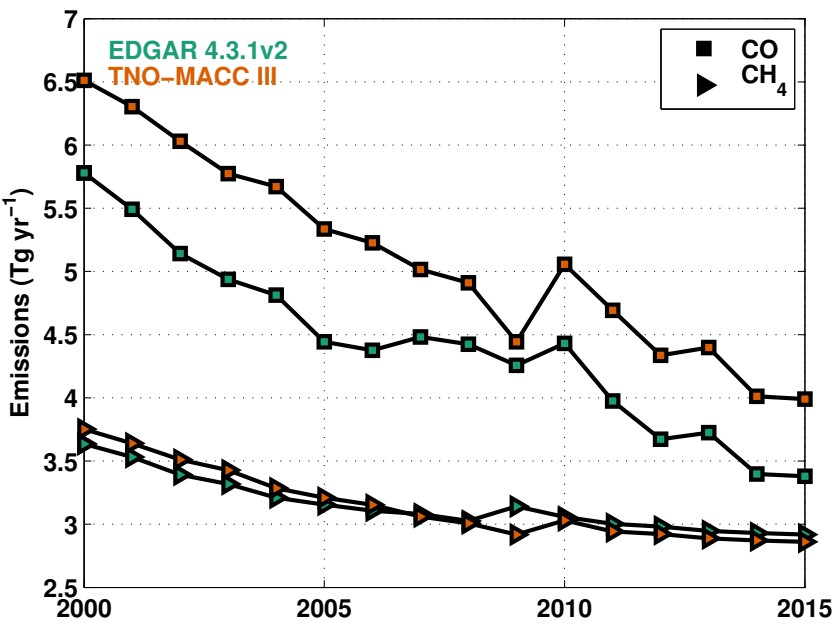

**Figure 3.** This figure shows the summed EDGAR (green) and TNO-MACC_III (orange) emissions within the study area for CO (squares), and $CH_4$ (triangles). The study area is defined in Figure 1. All emissions are shown in units of $Tg \cdot yr^{-1}$. Extrapolation begins after 2010 for EDGAR and 2011 for TNO-MACC_III.

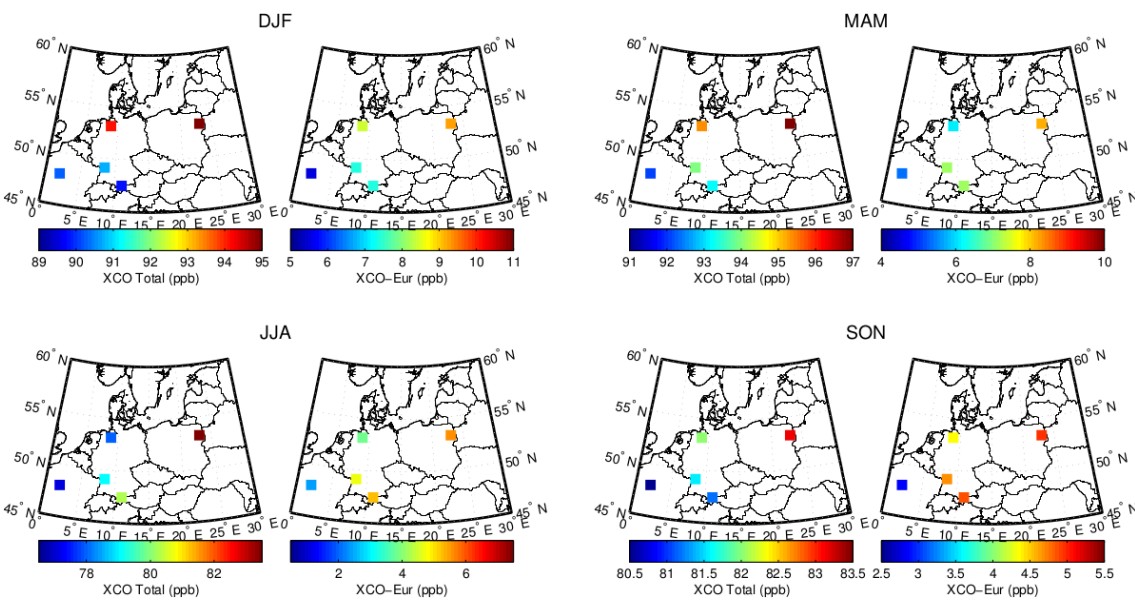

**Figure 4.** This figure compares seasonally-averaged modeled total $X_{CO}$ with the $X_{CO}$ contribution from emissions in Europe. Each season has two maps: the left map shows the total $X_{CO}$ and the right map shows the contribution from European emissions ($X_{CO-Eur}$). The spatial pattern of the gradients in modeled $X_{CO}$ between the TCCON stations is reflected in the European contribution.

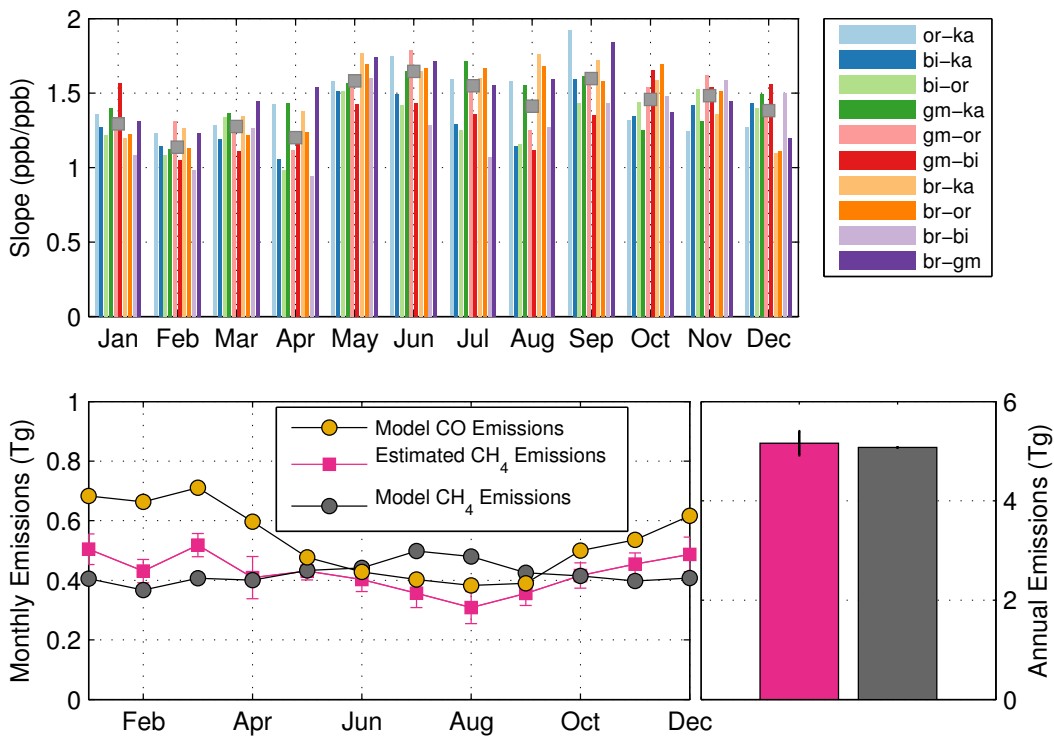

**Figure 5.** This figure shows the results from the modeling experiment using GEOS-Chem. The top panel shows the model $\Delta X_{CH_4}$–$\Delta X_{CO}$ slopes for each month and pair of stations (indicated by the colours). The median slopes for each month are overlaid in grey squares. The bottom left panel shows the model carbon monoxide emissions (excluding VOC and methane oxidation) and the model methane emissions. The inferred methane emissions from our tracer-tracer slope method are plotted in pink squares. The bottom right panel shows the annual methane emissions from the tracer-tracer slope method and the model.

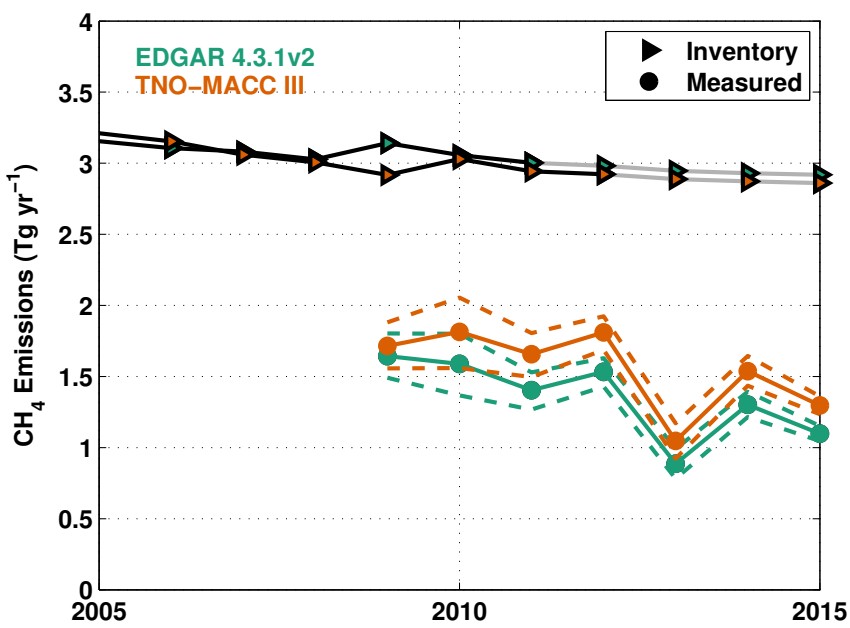

**Figure 6.** The black line is the summed EDGAR (green) and TNO-MACC_III (orange) methane emissions within the study area shown in Figure 1. The grey lines indicate the projected emissions based on scaling the country-level emissions reported by the UNFCCC (UNFCCC, 2017) to the area emissions in 2010 for EDGAR and 2011 for TNO-MACC_III. The lower solid lines show the emissions inferred from the TCCON anomaly analysis using CO emissions from the two models, and the dashed lines indicate the $5^{th}$ and $95^{th}$ percentiles.

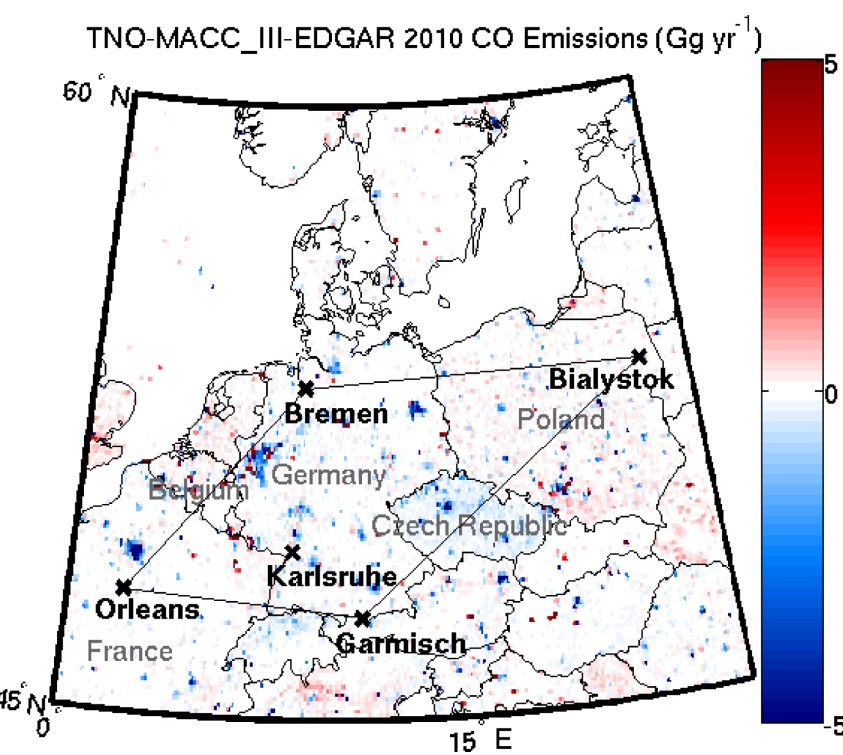

**Figure 7.** This map shows the difference between the TNO-MACC_III carbon monoxide emissions and the EDGAR emissions for the year 2010. The black straight lines delineate the study area from the surrounding region. The TCCON stations included in this study are marked with black "**x**" symbols and labeled in black bold font. The countries intersected by or contained within the study area are labeled in grey. Warm (red) colours indicate that the TNO-MACC_III inventory is larger than the EDGAR inventory; cool (blue) colours indicate that the EDGAR inventory is larger than TNO-MACC_III.

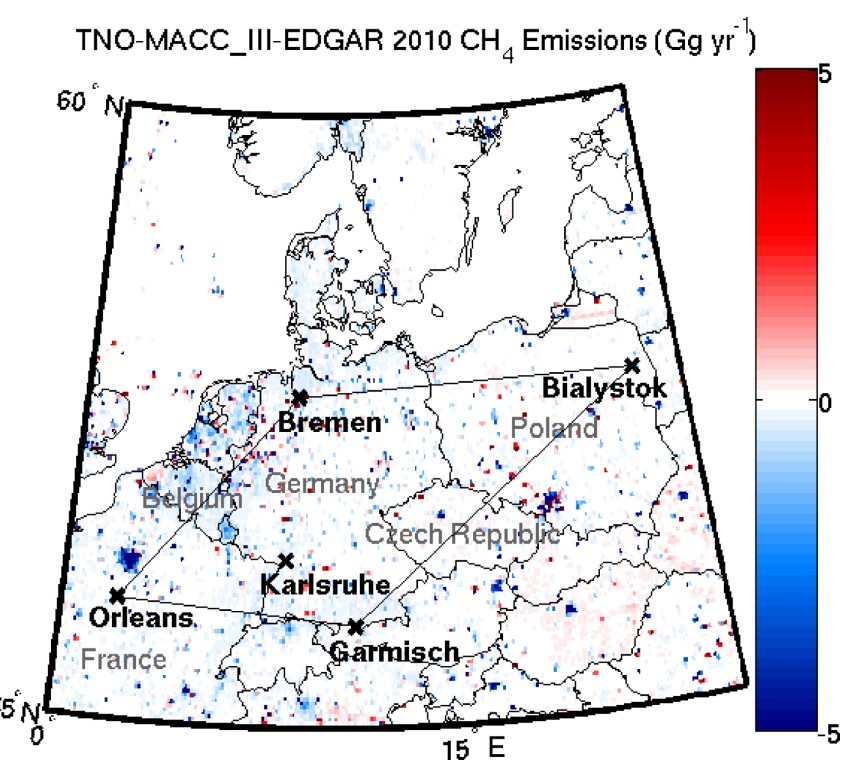

**Figure 8.** This map shows the difference between the TNO-MACC_III methane emissions and the EDGAR emissions for the year 2010. The labeling and colouring follows that in Fig. 7.

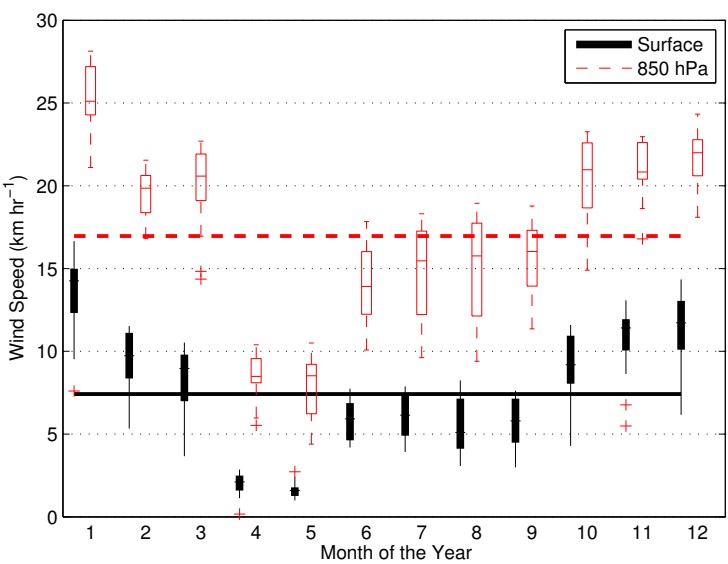

**Figure A1.** These boxplots show the NCEP/NCAR reanalysis long-term climatological monthly mean wind speeds at the surface (filled black boxes) and at 850 hPa (open red boxes) in the study area (see Figure 1, 7, or 8 for study area maps). The solid black and dashed red horizontal lines indicate the annual mean wind speed at the surface and 850 hPa (~1.5 km), respectively. Wind speeds aloft (on average 17 km $\cdot$ hr$^{-1}$) are significantly swifter than those at the surface (on average 7.5 km $\cdot$ hr$^{-1}$).

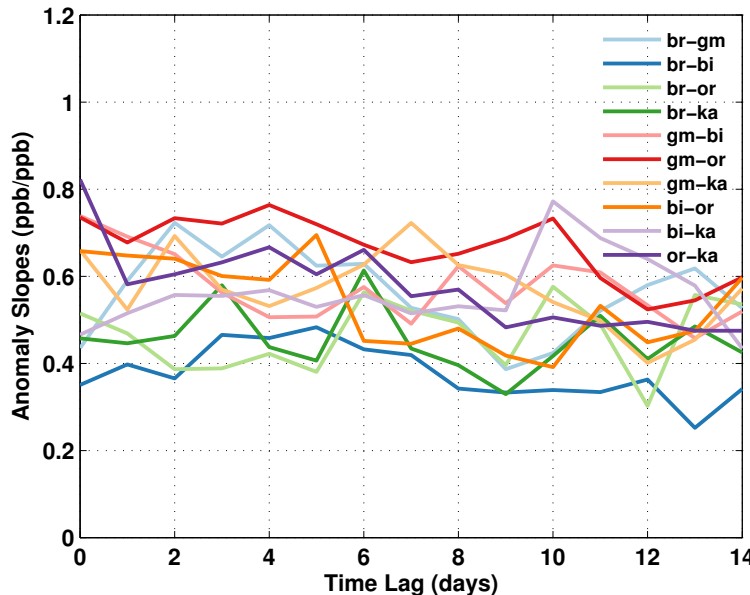

**Figure A2.** These are the anomaly slopes ($\Delta CH_4/\Delta CO$) in $\mathrm{ppb\,ppb}^{-1}$ for each station pair, for the entire time series. The anomalies are computed by subtracting data within the same SZA bin between two TCCON stations. For more detail, see §2 in the main text. The x-axis indicates the number of days separating the measurements. The legend identifiers are as follows: br - Bremen, gm - Garmisch, bi - Białystok, or - Orléans, ka - Karlsruhe.

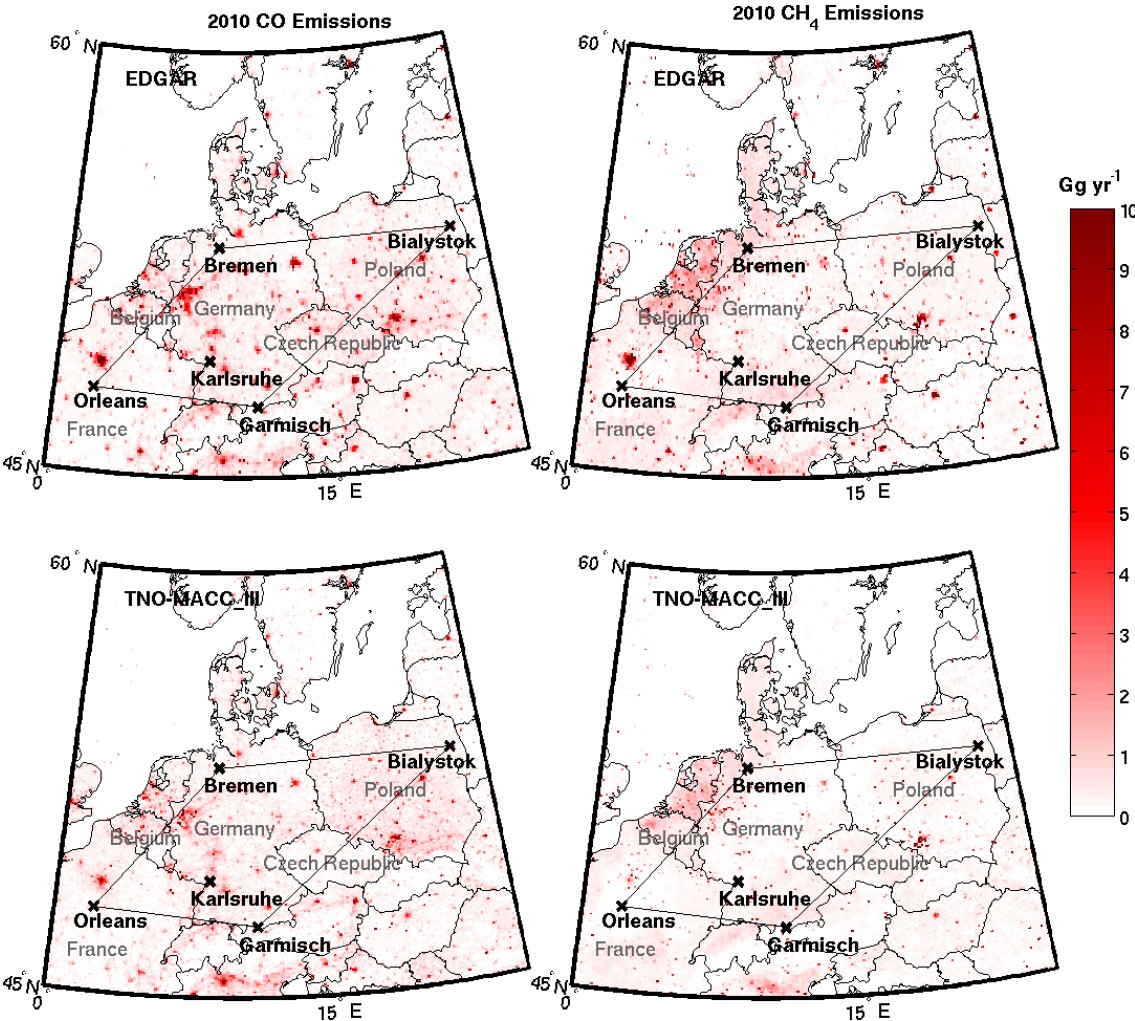

**Figure A3.** These maps show the inventory emissions for the year 2010 in the study area (delineated by the solid straight lines) and the surrounding region. The TCCON stations are marked with black "**x**" symbols and labeled in black bold font. The countries intersected by, or contained within, the study area are labeled in grey. The top left map shows the EDGAR v4.3.1 emissions inventory for carbon monoxide. The top right map shows the EDGAR FT2010 emissions inventory for methane. The lower left map shows the TNO-MACC_III emissions inventory for carbon monoxide. The lower right map shows the TNO-MACC_III emissions inventory for methane.

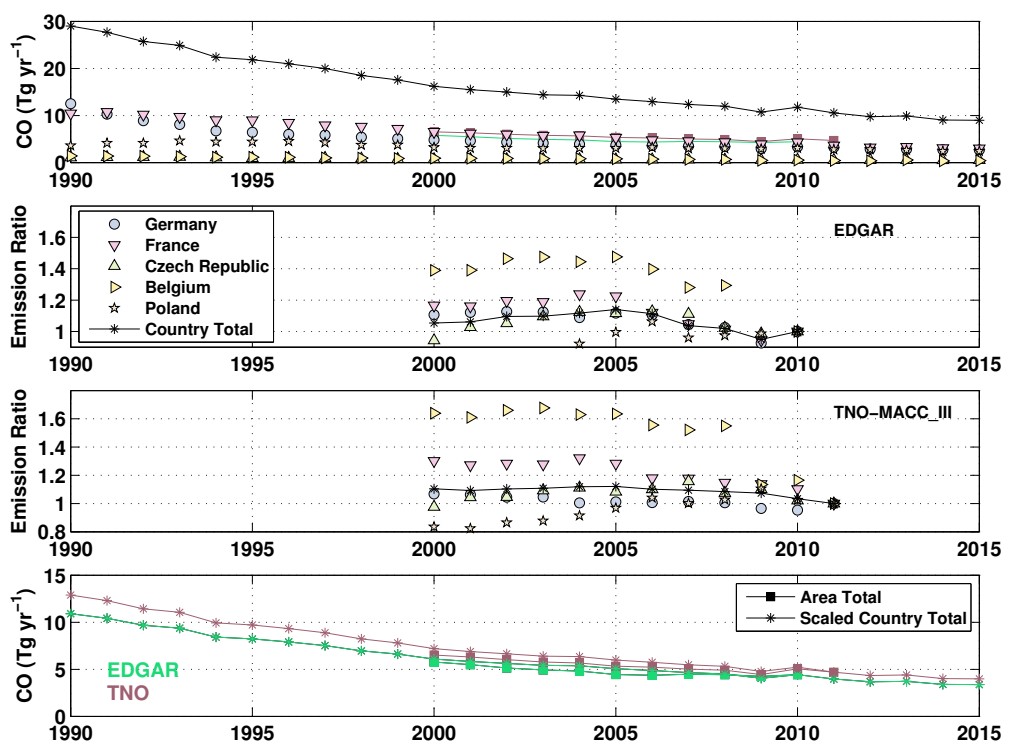

**Figure A4.** This four-panel plot shows the methodology for scaling the country-level reported emissions of CO to extrapolate the gridded inventory emissions to 2015. The top panel shows the CO emissions reported by the European Environment Agency (EEA) for the countries contained within the study area (Germany, France, Czech Republic, Belgium, Luxembourg, and Poland). The black stars with a joining line represent the summed total from the five countries. The EDGAR (green) and TNO-MACC_III (orange) inventories summed within the study area are plotted with squares joined by solid lines. The second panel shows the ratio between the individual country totals and the EDGAR area total, normalized to produce an emission ratio of 1 in 2010. The quantity with the least interannual variability in the ratio is from the country total (black stars with line). The third panel shows the ratio between the individual country totals and the TNO-MACC_III area total, normalized to produce an emission ratio of 1 in 2011. The quantity with the least interannual variability in the ratio is, again, from the country total. The bottom panel shows the scaled country total, normalized to produce the EDGAR CO emissions for 2010 and the TNO-MACC_III CO emissions for 2011. This permits us to compute a sensible emission for the study area through to 2015.

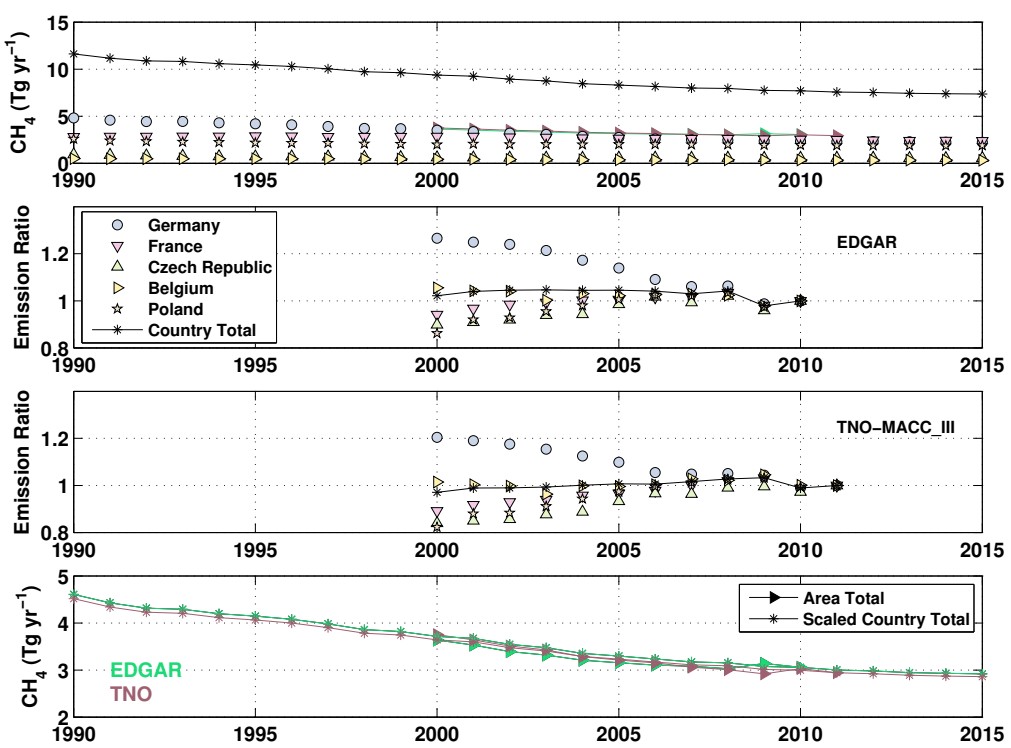

**Figure A5.** This four-panel plot shows the methodology for scaling the country-level emissions of $CH_4$ reported to the UNFCCC to extrapolate the gridded inventory emissions to 2015. The panels and symbols follow the same description as in Fig. A4.