# Peer review of "Emissions of methane in Europe inferred by total column measurements"

_Atmospheric Chemistry and Physics, 2018_

## Referee Comment (RC1) · Anonymous Referee #1 · 4 Jun 2018

The paper by Wunch et al. attempts to estimate methane ($CH_4$) emissions in Europe from a network of five ground-based sun-viewing total column spectrometers operated within the TCCON (Total Column Observing Network). The approach builds on calculating pair-wise differences between the (detrended) $CH_4$ and carbon monoxide (CO) concentrations at the five observatories and assuming that $CH_4$/CO emissions ratios scale like the detected concentration anomalies and that the input CO inventory is accurate. The authors conclude that two state-of-the-art $CH_4$ emissions inventories overestimate emissions in Central Europe.

I have major concerns detailed below which make me doubt that the results are robust and publishable in ACP. If I misunderstood things, please consider addressing these

points in greater depth by the (currently quite short) manuscript:

1. By design of the method, the $CH_4$ emissions estimate cannot be more accurate than the CO inventory used for scaling. A priori, I would argue that $CH_4$ inventories are per se better than CO inventories since there is less variability in $CH_4$ sources and less changes in source patterns. Figure 3 also indicates that the EDGAR and the TNO inventories show much larger discrepancies for CO than for $CH_4$ (likewise text p.6, l.21 ff). Then, the authors need to extrapolate the CO inventories under changing background conditions (change of source type) inducing further uncertainties. What is the total CO-induced error on your $CH_4$ emission estimates? Are the conclusions robust against this error? The conclusion needs to be quantitative in that respect.

2. The $CH_4$ and CO concentration anomalies are calculated pair-wise among the five stations. The ratio of the concentration anomalies is then used to scale the CO emission inventories to yield $CH_4$ emissions estimates. This requires that the concentration anomalies are representative of emissions inbetween the stations. In my understanding this requires that $CH_4$ and CO are emitted (roughly) coincident in time and space and that they share transport between stations. What is the temporal correlation of $CH_4$ and CO sources? What is the spatial overlap of $CH_4$ and CO sources (quantitatively)? Does CO (e.g. in the urban boundary layer) survive sufficiently long such that loss of CO over, say, 5-10 days is entirely negligible?

3. The analysis is based on five stations in Central Europe. As stated above, the analysis requires that the pair-wise anomalies are representative of emissions inbetween the stations. In my understanding, this implies that airmasses passing an upwind station should (at least for a large fraction of days) pass a downwind station. Is the network sufficiently dense to avoid a significant number of upwind airmasses being never sampled by a downwind station?

In fact, the time-lag analysis (Fig. A2, calulcating pair-wise anomalies not for the same day but for a time lag of several days) does not show any distinct dependence on time lag. This means that there is no indication that airmasses passing upwind stations also pass downwind stations (more often than sporadically).

Likewise for being representative of the entire Central European area (incircled by the stations), the five stations would need to sample (on a frequent basis) air masses that actually sampled the entire area. Is this condition met?

Is there evidence that these sampling issues are not significant?

The above concerns a touched on by the manuscript but I would argue that they are so critical that they require quantitative answers not just handwaving arguments. In general, my concerns 2 and 3 could be addressed by a comprehensive simulation study based on $CH_4$ and CO emissions inventories, a chemical transport model and an observation operator that mimicks the sampling by the ground-based spectrometers. I would recommend implementing such an approach to invalidate my concerns and to verifiy the assumptions or to reconsider the methodology.

---

## Referee Comment (RC2) · Anonymous Referee #2 · 23 Jul 2018

General comments:

This paper presents a study to extract regional methane emission information from long-term, stationary, ground-based solar absorption measurements of atmospheric total columns. The observations at the five selected sites are part of the Total Carbon Column Observing Network (TCCON) and are used for verifying annual methane emissions of two state-of-the art inventories EDGAR v4.2FT2010 and TNO-MACC_III. Anomalies are computed for methane and carbon monoxide between observations at two TCCON stations considering abundances measured at the same solar zenith angle and solar azimuth angle on the same day. The pair-wise anomalies between the TCCON stations are used to calculate the slope (CH4/CO) and then infer the methane emissions considering that the carbon monoxide emissions are well known. The au-

thors conclude that both the inventories significantly overestimate methane emissions and point to a large uncertainty in the spatial distribution of country level emissions of methane. Finally, the authors highlight the importance of long-term monitoring of total column measurements and simultaneous measurements of multiple atmospheric constituents.

I have some concerns which are mentioned below in the major and minor comments section. I recommend the publication of the manuscript after these points are addressed.

Major comments:

The paper relies on a set of five stations which are distributed far apart from each other and assumes that the CO and CH4 emitted between the stations are completely measured. – Can the authors show some evidence that for the majority of the days when the airmass was sampled by an upwind station then it was a few days later also sampled by the downwind station? It also assumes that the typical emissions are consistent over time periods longer than a few days so that the CO and CH4 are advected together. – This can only be true considering that no CO is lost, is that true especially as the major source of CO in Europe is coming from the urban area? As the author points out the TNO-MACC_III CO emissions are on average 15% higher and CH4 emissions are about 2% lower than the EDGAR v4.3.1 emissions in the study area. This indicates that the variability of CO is much higher for the selected study area and its surroundings. How does this variability propagate into the emission estimates of CH4? The author partly correlates the dip in CH4 emissions in 2013 shown in Fig. 14 to the small up-tick in CO in the same year. However, there is a similar CO up-tick in 2010 but no CH4 dip can be seen in the measurement. – Any explanation?

Minor comments:

Page 4 line 6-7: but each slope for each season has

Page 4 line 13: is the reference to the fig 4 correct?

Page 5 line 33: check your argument – if winter is cold it should rather result in increased heating needs

Page 8 line 29: reference of the European Environment Agency National Database (?) missing

---

## Author Comment (AC1) · 1 Feb 2019

Response to Anonymous Referee #1

We thank the referee for the thoughtful comments. We've responded to the major comments interspersed below. Referee comments are in red italics, and our responses are in black.

*By design of the method, the CH4 emissions estimate cannot be more accurate than the CO inventory used for scaling. A priori, I would argue that CH4 inventories are per se better than CO inventories since there is less variability in CH4 sources and less changes in source patterns. Figure 3 also indicates that the EDGAR and the TNO inventories show much larger discrepancies for CO than for CH4 (likewise text p.6, l.21 ff). Then, the authors need to extrapolate the CO inventories under changing background conditions (change of source type) inducing further uncertainties. What is the total CO-induced error on your CH4 emission estimates? Are the conclusions robust against this error? The conclusion needs to be quantitative in that respect.*

We agree completely that this method hinges on having an accurate CO inventory, and we have stressed this point further in the updated manuscript, and added a more quantitative discussion regarding the impact the CO inventory has on the methane emissions inferred.

In other regions of the world, in particular the US, other research has suggested that fugitive natural gas and agriculture emissions are significantly underestimated, and CO inventories are reasonably accurate (e.g., *Maasakkers et al.*, 2016; *Wunch et al.*, 2009; *Kort et al.*, 2016; *Frankenberg et al.*, 2016). We were surprised by our results from this region that seem to indicate that the methane emissions are overestimated.

*The above concerns a touched on by the manuscript but I would argue that they are so critical that they require quantitative answers not just handwaving arguments. In general, my concerns 2 and 3 could be addressed by a comprehensive simulation study based on CH4 and CO emissions inventories, a chemical transport model and an observation operator that mimicks the sampling by the ground-based spectrometers. I would recommend implementing such an approach to invalidate my concerns and to verifiy the assumptions or to reconsider the methodology.*

We have run GEOS-Chem v12.1.0 with EMEP CO and EDGAR $CH_4$ emissions for the year 2010. From this global 3D run, we have integrated the profiles of CO and $CH_4$, and computed dry-air mole fractions. The dry-air mole fractions were interpolated to the values at the locations of the 5 TCCON stations (Fig 1).

[Figure]

*Figure 1: XCH₄ (top panel) and XCO (bottom panel) integrated from GEOS-Chem and interpolated to the TCCON station locations.*

Pairwise differences at each hour were computed between the 5 stations (Fig 2), and the changes, or anomalies in CO were related to the anomalies in CH₄ (Fig 3).

[Figure]

*Figure 2: Pairwise differences are computed for XCH₄ (top panel) and XCO (bottom panel) between each pair of stations. These are the "anomalies."*

[Figure]

*Figure 3: These are the anomaly slopes per station pair, coloured by month of the year.*

Using these tracer-tracer anomaly slopes, we multiply the slope by the emission of CO from the model within the study region. We were able to return the annual $CH_4$ emissions in the model to within 2% of the true value (Fig 4).

[Figure]

*Figure 4: The top panel shows the slopes as a function of the month. The bottom left panel shows the monthly emissions from the model (CO and $CH_4$) and the inferred monthly methane emissions from the anomaly analysis. The bottom right panel shows the annual inferred methane emissions (pink) and the model emissions (grey).*

These results appear to show that our method is reasonably sound. The description and results from this

model experiment have been added in a new section to the paper.

Response to Anonymous Referee #2

We thank the referee for the thoughtful comments. We've responded to the major comments interspersed below. Referee comments are in red italics, and our responses are in black.

*The paper relies on a set of five stations which are distributed far apart from each other and assumes that the CO and CH4 emitted between the stations are completely measured. – Can the authors show some evidence that for the majority of the days when the airmass was sampled by an upwind station then it was a few days later also sampled by the downwind station?*

The footprint of a TCCON measurement is large – typically hundreds to thousands of kilometers. Thus, on average, measurements from each TCCON station will be sensitive to the entire region. To prove this more quantitatively, we have run the GEOS-Chem model, sampling the model at the TCCON stations and applying the same analysis. The results show that our anomaly slope method returns the model methane emissions to within 2%. We have added a new section to the paper describing this.

*It also assumes that the typical emissions are consistent over time periods longer than a few days so that the CO and CH4 are advected together. – This can only be true considering that no CO is lost, is that true especially as the major source of CO in Europe is coming from the urban area?*

We agree that our assumption is that the emissions are consistent over periods longer than a few days. The atmospheric lifetime of CO is about a month or so – much longer than the time it would take for the CO to advect from one end of Europe to the other. Although sources of CO and $CH_4$ are not necessarily (fully) co-located, with CO more in urban areas and $CH_4$ more in rural areas, in the European landscape these are relatively close to each other, and at scales of 10 x 10 or 25 x 25 km would appear mixed. We're unclear as to the referee's point regarding the urban source of CO – perhaps this is referring to co-emissions of $NO_x$ and $O_3$ in urban centres increasing OH? If so, we expect that a resulting shortened CO lifetime will still be long enough to advect across the study area

*As the author points out the TNO-MACC_III CO emissions are on average 15% higher and CH4 emissions are about 2% lower than the EDGAR v4.3.1 emissions in the study area. This indicates that the variability of CO is much higher for the selected study area and its surroundings. How does this variability propagate into the emission estimates of CH4?*

The relationship between CO and $CH_4$ should only rest on whether they are advected together, not as much on whether they are emitted in exactly the same place. Of course, in general, CO and $CH_4$ are **not** emitted from the same stack, and so it should be expected that the spatial distribution of CO and $CH_4$ are not identical. However, the emissions inventory maps show that typically the emissions are broadly co-located (i.e., within a few hundred km of each other).

*The author partly correlates the dip in CH4 emissions in 2013 shown in Fig. 4 to the small up-tick in CO in the same year. However, there is a similar CO up-tick in 2010 but no CH4 dip can be seen in the measurement. – Any explanation?*

The reviewer is correct that although the dip in 2013 is remarkable, there is also a CO uptick in 2010 without an impact on $CH_4$. Hence, we have removed the explanation for 2013 from the manuscript since we cannot also explain the lack of response for 2010.

*Minor comments:*
*Page 4 line 6-7: but each slope for each season has*

Fixed.

*Page 4 line 13: is the reference to the fig 4 correct?*

Indeed, no. That should be Fig. A3.

*Page 5 line 33: check your argument – if winter is cold it should rather result in increased heating needs*

Thank you for pointing this out. 2013 was cold, which would indicate that more CO would be produced due to heating needs (not less $CH_4$ consumption). This discussion was removed from the manuscript as mentioned earlier.

*Page 8 line 29: reference of the European Environment Agency National Database (?) missing*

Fixed.

[revised manuscript text omitted]

---

## Author Response (AR2)

Referee #1 Response

These are the referee's comments:

*The authors considered my concerns from round 1 of review. Concern 1 (quantitative assessment of CO error propagation) is now covered in sufficient detail.*

*Concerns 2 (spatial and termporal correlation of CO and CH4 emissions) and 3 (representativeness of station-gradients for the region) are addressed by a modelling study for the year 2010. The outcome of the modeling as reported is surprising (to me). It appears sufficient to put 5 sun-viewing spectrometers (in a more or less uncoordinated way) across Europe to estimate annual area-wide CH4 emissions with an error of 2%. I.e. the errors coming from imperfect representativeness and imperfect correlation of sources are negligible. CO lifetime issues are not addressed by the modelling study but they are argued to be negligible.*

*On the monthly and seasonal scale, the modelling indicates that the emission estimates are far off the truth (Fig. 4). The phases of the CH4 emissions seasonal cycle are even opposite in the true model and in the estimates. Differences generally amount to 20%. The errors cancel when aggregating over the whole year.*

*I do not understand these opposing results for the yearly and the sub-yearly time-scales. Is the seemingly great performance for annual emission estimates fortuitous? Is the year 2010 representative? Does the model analysis actually show that the station-gradients are sensitive to/representative for European emissions? Would it be thinkable that, for a full year, they are sensitive to the global / hemispheric scale (e.g. through the general atmospheric growth rate) and it happens that (e.g. for this particular year) the „European growth rate" is just the same as the global/hemispheric one?*

*The manuscript would benefit from some more detailed discussion along the above questions.*

We have addressed the referee's concerns by adding an additional Figure (new Fig. 4) that shows the sensitivity of the XCO to European emissions from a tagged-tracer run of the model. We have also added text in Section 2.2 to describe and clarify the results of the sensitivity test. Finally, we have further stressed the discrepancy between the seasonal and annual performance of the model and suggested a mechanism that may explain it.

The tracked-changes version of the updated manuscript begins on the next page.

[revised manuscript text omitted]